# Novel *Salmonella* Phage, vB_Sen_STGO-35-1, Characterization and Evaluation in Chicken Meat

**DOI:** 10.3390/microorganisms10030606

**Published:** 2022-03-12

**Authors:** Dácil Rivera, Andrea I. Moreno-Switt, Thomas G. Denes, Lauren K. Hudson, Tracey L. Peters, Reham Samir, Ramy K. Aziz, Jean-Paul Noben, Jeroen Wagemans, Fernando Dueñas

**Affiliations:** 1Escuela de Medicina Veterinaria, Facultad de Ciencias de la Vida, Universidad Andres Bello, Santiago 8320000, Chile; ferdu26@gmail.com; 2Millennium Initiative for Collaborative Research on Bacterial Resistance (MICROB-R), Santiago 7550000, Chile; andrea.moreno@uc.cl; 3Escuela de Medicina Veterinaria, Facultad de Agronomía e Ingeniería Forestal, Facultad de Ciencias Biológicas, Facultad de Medicina, Pontificia Universidad Católica de Chile, Santiago 7810000, Chile; 4Department of Food Science, University of Tennessee, Knoxville, TN 37996, USA; tdenes@utk.edu (T.G.D.); lkhudson@utk.edu (L.K.H.); tpeter21@vols.utk.edu (T.L.P.); 5Department of Microbiology and Immunology, Faculty of Pharmacy, Cairo University, 11562 Cairo, Egypt; reham.samer@pharma.cu.edu.eg (R.S.); ramy.aziz@gmail.com (R.K.A.); 6Microbiology and Immunology Research Program, Children’s Cancer Hospital Egypt 57357, 11617 Cairo, Egypt; 7Biomedical Research Institute and Transnational University Limburg, Hasselt University, Agoralaan D, 3590 Hasselt, Belgium; jeanpaul.noben@uhasselt.be; 8Department of Biosystems, KU Leuven, 3001 Leuven, Belgium; jeroen.wagemans@kuleuven.be

**Keywords:** *Salmonella Enteritidis*, *Salmonella*–phage in food, *Siphoviridae*, *siphoviral morphotype*, receptor-binding proteins

## Abstract

Salmonellosis is one of the most frequently reported zoonotic foodborne diseases worldwide, and poultry is the most important reservoir of *Salmonella enterica* serovar Enteritidis. The use of lytic bacteriophages (phages) to reduce foodborne pathogens has emerged as a promising biocontrol intervention for *Salmonella* spp. Here, we describe and evaluate the newly isolated *Salmonella* phage STGO-35-1, including: (i) genomic and phenotypic characterization, (ii) an analysis of the reduction of *Salmonella* in chicken meat, and (iii) genome plasticity testing. Phage STGO-35-1 represents an unclassified siphovirus, with a length of 47,483 bp, a G + C content of 46.5%, a headful strategy of packaging, and a virulent lifestyle. Phage STGO-35-1 reduced *S. Enteritidis* counts in chicken meat by 2.5 orders of magnitude at 4 °C. We identified two receptor-binding proteins with affinity to LPS, and their encoding genes showed plasticity during an exposure assay. Phenotypic, proteomic, and genomic characteristics of STGO-35-1, as well as the *Salmonella* reduction in chicken meat, support the potential use of STGO-35-1 as a targeted biocontrol agent against *S. Enteritidis* in chicken meat. Additionally, computational analysis and a short exposure time assay allowed us to predict the plasticity of genes encoding putative receptor-binding proteins.

## 1. Introduction

Salmonellosis is one of the most frequently reported zoonotic foodborne diseases [1,2]. The causative agent, *Salmonella*, can be transmitted to humans along the farm-to-fork continuum, commonly through contaminated foods of animal origin [3]. *Salmonella* spp. are estimated to cause 93.8 million cases of acute gastroenteritis and 155,000 deaths globally [4]. The Centers for Disease Control and Prevention (CDC) estimates that *Salmonella* spp. cause 1.2 million illnesses, 23,000 hospitalizations, and 450 deaths in the United States each year, resulting in an estimated USD 400 million loss in direct medical costs [2]. In Europe, during 2016, *Salmonella* spp. were responsible for 24.4% of all foodborne outbreaks [1,2], while they caused 43% of foodborne outbreaks in the United States in 2018 [3].

*Salmonella* is classified into two species, *S. enterica* and *S. bongori*. *S. enterica* consists of six subspecies: *enterica*, *salamae*, *arizonae*, *diarizonae*, *houtenae*, and *indica*, with more than 2600 serovars, the majority of which can cause infections in animals and humans [5]. Among the most frequent disease-causing serovars are *Salmonella* Enteritidis, Typhimurium, Heidelberg, and Newport [6]. These epidemiologically important serovars are related to a high rate of foodborne salmonellosis outbreaks in humans. Together, these serovars are considered to be the third highest cause of human death among diarrheal diseases worldwide [6]. The emergence of *S. Enteritidis* was first noted in the 1980s and has increased over time [7]. In 1995, 36% of worldwide foodborne outbreaks were attributed to *S. Enteritidis*, compared to 65% in 2002 [7]. The main source of *S. Enteritidis* is the consumption of animal foodstuffs, such as eggs and poultry meat and their derivatives, since *S. Enteritidis* can persist in the intestinal tract of chickens, thereby creating chronic or asymptomatic carriers that continue to excrete *S. Enteritidis* in their feces [6]. Therefore, it is difficult to control *Salmonella* owing to its ability to remain in animal production systems and the surrounding environment [6]. To reduce the incidence of salmonellosis, it is necessary to develop new mitigation strategies to control and reduce persistent *Salmonella* spp. in animal production systems [8].

Bacteriophages (hereafter referred to as “phages”) are a promising alternative to traditional food safety preservation approaches [9], especially in view of their activity against antimicrobial-resistant bacteria [10]. Phage-based interventions in food were shown to decrease the loads of pathogenic bacteria, such as *Listeria monocytogenes*, *Salmonella*, and *Campylobacter jejuni*, as well as spoilage microorganisms in fruits, dairy products, poultry, and red meats [9,11]. Several approaches for using phages to control *Salmonella* at the farm level have been studied. For instance, the use of phages to prevent *Salmonella* colonization in animals was previously investigated in in vivo models, including birds and pigs [12]. *Salmonella* phages have also been successfully applied as a biocontrol tool in chicken meats, with reductions of viable counts of *S. Enteritidis* and *S. Typhimurium* in the phage-treated samples at 3.06 and 2.21 log CFU/piece, respectively (*p* < 0.05) [13].

While promising, phages are antimicrobials that evolve during phage–host interactions, in which the bacterium could become phage-resistant and the phage could evolve to develop counter resistance [14,15]. Bacteria–phage interactions are complex and several factors, such as bacterial aging, decrease in lytic efficiency, or lysis inhibition, have been associated with changes in the lysis plaque morphology [16]. Moreover, phage sub-isolates (genetic variants) could be selected under laboratory conditions [14,15,17,18]. Laboratory exposure of the bacterial host to phages, followed by subsequent phenotypic and genomic characterization, represent assays that could identify possible changes in receptor-binding proteins (RBPs) as consequence of coevolution events [14,15]. These changes could further influence the ability to recognize the hosts and to overcome phage resistance [14,15,17,18]. Although phage variants have been reported, the characteristics of their emergence and their effect on phage applications have not been comprehensively explored.

Phages stand out for being highly abundant in the environment, in animal production systems [19], and in human fecal samples [20], but there is considerable difficulty in their classification [21]. One of the most abundant phage morphology-based families is the classical *Siphoviridae (siphoviral morphotype)*, which has been reclassified most recently into different genome-based taxa [22]. Phages belonging to the siphovirus morphotype are characterized by long flexible tails [23]. The phage heads and tails are assembled separately, and the tails are built of stacked disks of six subunits [23].

Currently, 26% of the bacteriophage genomes in the National Center for Biotechnology Information (NCBI) RefSeq database (https://www.ncbi.nlm.nih.gov/labs/virus/vssi/#/, accessed on 10 March 2021) correspond to unclassified *Siphoviridae* (2775 of 10,525 genomes). Of these, 14% (60/425), correspond to unclassified *Siphoviridae* that infect the *Salmonella* genus. Therefore, it is of high importance to characterize new siphoviruses with the perspective of using them for the biocontrol of *Salmonella* [11]. To this end, comparative genomics have the potential to strongly increase our understanding of phage diversity and function [24] and of genome packaging strategies [25,26]. Significantly, lifestyle identification is critical for determining the role of individual phage species within ecosystems, their effect on host evolution, and their safety for use in biocontrol. Lifestyle classification includes temperate and virulent categories [27,28]. Temperate phages have high genomic plasticity, which may involve the mechanisms of gene flows caused by a recombination of the genome concatemers that are packaged into the virion [25,26]. Aditionally, the modular organization of temperate phage genomes, recognized as a mosaic of genomic regions, with similar sequences within pairs of dissimilar genomes, drive gene exchange between phages of different phylogenetic origins [27]. This gene flow tends to occur between recombinases, transposases, and nonhomologous end joining, suggesting that both homologous and generalized recombination contribute to gene flow [27]. On the other hand, it should be noted that there are phages with low gene flux, which are usually virulent phages, with genes encoding functions involved in cell energetics, nucleotide metabolism, DNA packaging and injection, and virion assembly [27].

The packaging type corresponds to a powerful molecular motor, and it is composed of the portal protein (which provides a portal for DNA entry), the large terminase subunit whose ATPase activity drives DNA translocation, a small terminase subunit (which recognizes the viral packaging site) [29], and the action of headful nucleases (which cut the viral genomes before and after DNA packaging) [30,31,32]. Therefore, the phylogenetic origin of the phage, the type of terminase, and the packaging mechanisms should be related [26]. For example, representative termini of different phages have been described: 5’cos (lambda), 3’cos (HK97), *pac* (P1), head without *pac* site (T4), DTR (T7), and host fragment (Mu) [26]. Some *Salmonella* phages have been classified within the *Siphoviridae* family (currently *siphoviral morphotype)*, and they present a type of *pac* site-directed headful packaging mechanism, including *S. enterica*-phages 9NA, FSL_SP-062, FSL_SP-069, and Sasha, Sergei and Solent [33]. Standardization, especially for circularly permuted phages, will facilitate the comparison of phage genomes and the identification of homologs [26].

Siphoviral phage genome organization is modular and prone to horizontal gene exchange; nevertheless, related functionalities can be recognized between different modules across phage genomes [28]. The amino acid sequences in these gene products share conserved protein domains (CDDs), which contain common sequence patterns or motifs, characterized as functional and/or structural units in a polypeptide sequence [28]. In molecular evolution, these domains may have been used as building blocks, and may have been recombined in different arrangements to modulate protein function, so their analysis can derive important information about genetic plasticity in the phage genome [34,35,36]. In addition, this modular conformation is manifested in other adhesin-like tail genes, which have been presented in the podovirus *Salmonella* P22 and siphovirus. In these genes, a common genetic origin has been demonstrated, with minor mutations in the central segments that could impact the folding of the encoded proteins [37]. The aim of this study was to comprehensively characterize a newly isolated *Salmonella* phage, STGO-35-1 (vB_Sen_STGO-35-1), for which in silico genomic (via comparative genomics) and phenotypic analysis may identify promising and rapid approaches for better understanding phage-based biocontrol.

## 2. Materials and Methods

### 2.1. Isolation and Characterization of Phage vB_Sen-STGO-35-1

#### 2.1.1. Propagation Conditions

A *Salmonella* phage, which we named vB_Sen_STGO-35-1 (STGO-35-1), was isolated from a backyard chicken flock, using *Salmonella* Enteritidis strain DR028 as an isolating and propagating host [38]. The phage was isolated as previously described [38]. Briefly, the isolation consisted of the enrichment of 1 g of a cloacal swab in a 100 mL culture of four *S. enterica* serovars (Infantis, Heidelberg, Typhimurium, and Enteritidis) grown in tryptic soy broth (TSB, Becton-Dickinson, Franklin Lakes, NJ, USA). Cultures were diluted tenfold, incubated at 37 °C for 18 ± 2 h, and then centrifuged. The supernatants were filtered through 0.22 µm filters to generate a primary lysate. Subsequently, the lysate was mixed with each host individually, with 100 µL of phage lysate and 300 µL from a 1:10 *v/v* dilution of an exponentially growing culture of each serovar. The mixture was added to soft trypticase soy agar (TSA; 0.7% Becton-Dickinson, Franklin Lakes, NJ, USA), plated on TSA, and incubated for 18 ± 2 h at 37 °C. From that plate, one lysis plaque was selected to be purified with the host for at least three subsequent passages. For propagation, plaques with confluent lysis were flooded with 10 mL of sodium–magnesium (SM) buffer (50 mM Tris-HCl pH 7.5, 0.1 M NaCl, 0.01 mM MgSO_4_) and filtered (0.22 µm). This lysate of the original phage is referred to hereafter as the wild type of STGO-35-1.

The host range assay (Appendix A), for this phage, was performed using the double-layer agar method, as previously described [38], against 23 different *Salmonella* serovars. Briefly, the characterization of phage host range was performed by spotting 5 mL of phage lysates (approximately 3 × 10^5^ PFU/mL) on a host cell lawn prepared with a 1:10 dilution of an overnight culture of the host strains in 4 mL of soft agar (0.7% TSA). Plates were incubated for 16 to 18 h at 37 °C and then examined for lysis (either present or not present), considering that clean and turbid plaques indicate the presence of lysis and that the absence of plaques indicates no lysis. Experiments were performed in two independent replicates.

#### 2.1.2. One-Step Growth

A one-step growth curve was plotted according to a previously described standard protocol [39]. Briefly, *S. Enteritidis* was infected with the STGO-35-1 phage at a multiplicity of infection (MOI) of 0.01 and incubated at 37 °C. Then, after an incubation time of 10 min, two 100 µL samples were centrifuged at 13,500 rpm and were collected every 10 min. The supernatant was separated to determine both the viral titer (PFU/mL) and the bacterial cell count (CFU/mL). The burst size was calculated through the quantification of the infective centers (average three higher viral titers/three lower viral titers). The latency period was calculated as the mean between these time points (immediately post-lysis) and the previous time point (immediately pre-lysis). This assay was conducted in triplicate.

#### 2.1.3. Transduction Efficiencies

Transduction efficiencies were determined by estimating the frequency of transduction by quantifying the phage-mediated acquisition of a resistance gene on *S. Typhimurium* 14,028 s, as previously described [40]. The transduction frequency was calculated as the number of infectious centers CFU/PFU mL. Experiments were performed in duplicate.

#### 2.1.4. Microscopic Characterization of Phage vB_Sen-STGO-35-1

Transmission electron microscopy (TEM) was used to ascertain the morphological traits. For this, a first scan of the sample was carried out to measure the viral particles, as recommended by Ackermann [41]. Briefly, phage particles, which were previously purified and precipitated with polyethylene glycol PEG8000 (Sigma-Aldrich, St. Louis, MO, USA), were used and stored in SM buffer. Next, phage particles were washed with 0.1 M ammonium acetate and centrifuged at 21,000× *g* in a microcentrifuge (Thermo Fischer Scientific, Waltham, MA, USA) and were deposited onto 150–200 mesh carbon-coated Formvar film copper grids and stained with 1% phosphotungstic acid (PTA, pH 7.4) and imaged with a JEOL 1400 Flash TEM at magnifications of 50,000× to 100,000× at 85 kV. Images were analyzed using Fiji3 [42].

#### 2.1.5. Genomic and Phylogenetic Analyses of Phage vB_Sen-STGO-35-1

For genomic and taxonomic characterization, DNA was analyzed using the phenol–chloroform method, and then precipitated with ethanol, as previously described [38]. To eliminate exogenous genomic material, we treated phage stocks (titer > 5 × 10^10^ PFU/mL) with 2 mM CaCl2, 5 μg/mL DNase-I (Promega BioScience, Madison, USA) and 30 μg/mL RNase-A (Sigma-Aldrich, Darmstadt, Germany) for 30 min at room temperature. To inactive enzymes, we incubated samples at 65 °C for 10 min and then 2 mg/mL Proteinase K (Promega BioScience, Madison, MA, USA) was added. The rest of the Sambrook and Russel protocol was followed [43]. Then, DNA concentration and quality were determined using a Maestro Nano Pro-Spectrophotometer (Maestrogen Inc., Hsinchu, Taiwan), as previously described [15].

Sequencing libraries were prepared using the Nextera XT library preparation kit (Illumina, San Diego, CA, USA) and sequencing was conducted using the Illumina HiSeq platform at MicrobesNG (Birmingham, United Kingdom). Trimmomatic (v0.35; ILLUMINACLIP: NexteraPE-PE.fa:2:30:10 LEADING:3 TRAILING:3 SLIDINGWINDOW:4:15 MINLEN:36) [44] was used to trim raw reads and then FastQC (v0.11.7) [45] was used to assess quality. SPAdes (v3.12.0; careful option) [46] was used to assemble the trimmed reads and assembly statistics were generated using BBMap (v38.88) [47], SAMtools (v0.1.8) [48], and QUAST (v4.6.3) [49].

The assembled genome was re-oriented to begin at the large terminase subunit, and reads were mapped to the assembly to check that coverage across the new junction was consistent with the rest of the assembly. The re-oriented assembly was annotated with RASTtk (with the pipeline customized to run “annotate-proteins-phage” before “annotate-proteins-kmer-v2”) [50] and ARAGORN v1.2.41 was used identify tRNA genes [51]. A genome map of STGO-35-1 was generated with Artemis and DNAPlotter [52]; subsequently, lifestyle prediction (temperate or virulent) was conducted using BACterioPHage LIfestyle Predictor (BACPHLIP v0.9.6), which detects the presence of conserved domains and uses these data to predict lifestyle using a random forest classifier on a dataset of 634 phage genomes [24], and PhageTerm [25] was used to predict the packaging mechanism and the characteristics of the termini. This software uses raw reads of a phage sequenced with a sequencing technology using random fragmentation and its genomic reference sequence to determine the termini position, first segments the genome according to coverage using a regression tree.

In addition, the phage genome was compared to nucleotide sequences available from the GenBank-NIH database. This analysis was carried out with a complete alignment of the nucleotide sequences using BLASTn, and relatedness was evaluated using JSpeciesWS [53], which considered the best query score, highest identity (close to 100%), and best probability (e-value under 0.0). The similarity between sequences was plotted using EasyFig [54]. Subsequently, a phylogenetic analysis [55] was carried out using COBALT [56], with STGO 35-1 as the reference. A max seq difference (>0.5) of 0.75 was considered for this analysis. Additionally, a phylogenetic analysis based on the large terminase subunit was also conducted using COBALT [55,56]. The evolutionary distance between two sequences was modeled as the expected fraction of amino acid substitutions per site given the fraction of mismatched amino acids in the aligned region using the model proposed by [57]. The max seq difference used was 0.75, and the minimum distance to conform the groups was 0.01.

#### 2.1.6. Structural Proteome Analysis of Phage vB_Sen-STGO-35-1

Structural proteome analysis on STGO-35-1 phage particles was conducted using LC-MS/MS, as described previously by Wagemans et al. [58]. Briefly, phage lysate was centrifuged for 10 min at 13,000 rpm. Then, supernatant was filtered through 0.22 µm syringe-attached filters. This procedure was performed three times and, subsequently, the phage lysate was precipitated with polyethylene glycol PEG8000 (Sigma-Aldrich, St. Louis, MO, USA) and stored in SM buffer. Purified lysates were analyzed in the Department of Biosystems, KU Leuven, Belgium. Phage proteins were extracted from a PEG-purified phage stock (10^11^ PFU/mL) using a chloroform:water:methanol extraction (1:1:0.75). The protein pellet was resuspended in SDS-PAGE loading buffer (40% glycerol, 200 mM Tris-HCl pH 6.8, 4% sodium dodecyl sulphate (SDS), 0.4% bromophenol blue, 8 mM EDTA, 5% beta-mercaptoethanol and separated on a 12% SDS-PAGE gel. After Coomassie staining, gel slices were picked across the whole lane and processed for mass spectrometry analysis using LC-MS/MS on an Easy-nLC 1000 liquid chromatograph (Thermo Scientific), coupled to a mass calibrated LTQ-Orbitrap Velos Pro via a Nanospray Flexion source (Thermo Fisher Scientific) using sleeved 30 μm ID stainless steel emitters, as described previously by Shevchenko et al. [59] and Ceyssens et al. [60]. The raw data were analyzed using SEQUEST v1.4 (Thermo Fisher Scientific) and Mascot v2.5 (Matrix Sciences). The amino acid sequence analysis of phage structural proteins (putative proteins/functions), previously predicted with LC-MS/MS, was conducted with HHpred multiple sequence alignment [61] against Protein DataBank (PDB), uniProt, NCBI_CDD conserved_domain_database, and SCOpe [62]. Then, the amino acid sequence that presented the best probability, score, identity, and query was selected.

### 2.2. Reductions of S. Enteritidis in Chicken Meat Using STGO-35-1 Phage

#### 2.2.1. Ideal Multiplicity of Infections (MOIs) to Obtain Maximum Adsorption Rate

We first tested different MOIs. Briefly, *S. Enteritidis* cultures were grown to an OD_600_ of 0.5 and centrifuged at 8000 rpm for 5 min. Bacteria pellets were gently resuspended in 1 mL of TSB, mixed with the STGO-35-1 at various MOI (0.1, 1, 10, and 100) for incubation at 37 °C. Samples were collected at 0, 10, 20, 30, 40, 50, and 60 min and placed on ice until the final time point and then centrifuged for 4 min at 13,000 rpm. Next, the phages contained in the supernatant were serially diluted at 1:10 *v*/*v* in SM buffer and were titrated using the double layer agar method (incubated overnight at 37 °C). The adsorption rate was calculated according to the number of PFU/mL remaining in the supernatant and expressed as a percentage of the number of initial concentrations (PFU/mL) in the cultures [39].

#### 2.2.2. Assay in Chicken Meat

An assay to evaluate the reduction in *S. Enteritidis* in artificially contaminated chicken meat was conducted as previously described [13]. For this, a piece of 500 g of chicken meat was bought from local retail and transported to the laboratory under refrigerated conditions. Before the experiments, molecular screening was performed to confirm the absence of *S. enterica* and phages similar to STGO-35-1 in the chicken meat used for the assays. We use *InvA*-PCR, as previously described, to test for *Salmonella* presence [63]. Primers that targeted STGO-35-1 were designed with PrimerBLAST (https://www.ncbi.nlm.nih.gov/tools/primer-blast/, accessed on 9 March 2022): forward primer 5’ GGACGCGTAGCTTAATTGGT 3’ and reverse primer 5’GTGGACACGGACGGATTTGA 3’ of a tRNA gene of STGO-35-1 were used. Both PCR tests were conducted before the assays.

Chicken meat was prepared by cutting approximately 2 cm^2^ pieces, which were deposited on a Petri dish. The surface of each piece was inoculated by spotting 50 uL of a concentration of 4 × 10^6^ CFU/mL of a strain of *S*. Enteritidis resistant to nalidixic acid (*S*E*^nalr^*), which was then allowed to air dry for 30 min at room temperature under aseptic conditions. Subsequently, 50 uL of phage lysate was added to the surface of the meat pieces at 4 × 10^6^ PFU/mL (MOI = 1) and the cultures were cultivated as mentioned below. Two controls were prepared, one control with only 50 uL of *S*E*^nalr^* in the chicken meat pieces in the absence of phage, and another control with only the phage added in the absence of *S*E*^nal^*, and both controls were dried in the same conditions. Inoculated chicken meats were incubated at 4 °C for 7 days. Each day, samples were collected to determine bacterial concentrations and phage titers. For this, three different pieces of meat of approximately 2 g were used in each sampling time. Meat pieces for *Salmonella* and phage quantifications were placed in tubes with 5 mL of 0.8% NaCl, incubated at room temperature for 10 min, and mixed by shaking at 50 rpm. Samples were then centrifuged at 12,000 rpm for 30 min, and bacteria pellets were resuspended in 1 mL of TSB, and the mixture was further used to quantify the bacterial concentration using plate counts on nalidixic acid-added TSA plates, which was expressed as CFU/mL. The meat pieces were also resuspended in 5 mL of buffer SM and then separated from the supernatant, which was filtered with a 0.22 µm filter, following the addition of 1% chloroform; this was conducted to quantify the phage titer via spot-testing in double agar with the original host, which was expressed in PFU/mL. The experiment was repeated three times for each experimental group. The daily difference between the CFU/mL of *S. Enteritidis* in comparison to controls was statistically tested using an ANOVA test (with a *p* < 0.05 considered to be a significant difference). The statistical software Infostat was used for this analysis (released 2016: https://www.infostat.com.ar, accessed on 9 March 2022).

#### 2.2.3. Genome Stability Testing

To determine the stability of phage STGO-35-1, we followed two approaches: (i) an in silico analysis and prediction of RBPs putative proteins and (ii) an exposure assay followed by sequencing. For the in silico prediction of RBPs, we used amino acid sequences and structural proteome data, as described previously (Section 2.1.6). The proteins that represented the best prediction were compared with similar sequences via Clustal-O alignment [64]. From this, each similar sequence selected as the main RBP was analyzed through comparison to the database NCBI_CDD [62]. We analyzed putative conserved domains in three portions of the protein (amino terminal, central, and carboxy terminal). With this information, the most similar sequence was selected using a Markov model in the Modeler program [65]. Modeler provided the most probable three-dimensional structure of the protein and described the surface amino acidic residues available to interact with different molecules of interest (such as bacterial receptors). In addition, it provided information on the origin of the crystallized structure in a Protein Data Bank (PDB) format

For the exposure assay, both phage and *S. Enteritidis* were co-cultured at an MOI of 0.1, with a phage concentration of 4 × 10^7^ PFU/mL and *S. Enteritidis* at 4 × 10^8^ CFU/mL. This mixture was incubated for 60 min at 37 ° C, and then the mixture was blended with soft agar (TSA 0.7% Becton-Dickinson, Franklin Lakes, NJ, USA), and it was then inoculated on TSA (Becton-Dickinson, Franklin Lakes, NJ, USA) and incubated for 18 ± 2 h. The morphology of each lysis plaque was visually examined after one day. Three different plaque morphologies (P1, P2, and P3) were selected and propagated (as described in Section 2.1.1) for further comparative genomic analysis. For this purpose, phage DNA was purified and sequenced (following the previously described protocol, Section 2.1.5). Mutations were identified using McCortex (v.0.0.3) [66], pipeline (“vcfs” argument, links and joint calling, both bubble and breakpoint callers, and a kmer size of 81) with the wild-type phage assembly as the reference. The wild-type phage reads were also included as a control to account for spontaneous mutations. The genes containing mutations were further explored with InterProScan [67] and HMMER [68]. The mutations found in P1, P2, and P3 were classified by: (i) the effect of the mutation on amino acid substitution (based on charge and polarity, as described by Hanada et al. [69]; this included radical nonsynonymous substitution (RNS), synonymous substitution (SS), or conservative nonsynonymous substitution (CNS); and (ii) frequency as compared to frequency in control phage (wild type).

#### 2.2.4. Data Availability

The genomic sequences were deposited in NCBI databases under the following IDs: GenBank MW477799.1. (NC_054648.1), BioProject PRJNA691979, and BioSamples SAMN17570725 (P1), SAMN17570726 (P2), and SAMN17570727 (P3).

## 3. Results and Discussion

### 3.1. Phenotypic Characterization of STGO-35-1 Phage

STGO-35-1 was isolated on the *S. Enteritidis* strain DR028 (Appendix A), drawn from a backyard poultry flock in central Chile. The isolated phage generated clear lysis plaques, with an average plaque size of 2 mm, after three consecutive purifications with its *S. Enteritidis* host. The phage was purified, and a host range analysis of STGO-35-1 showed a narrow host range (Appendix A). The phage was capable of lysing 4 out of 23 tested strains, representing *Salmonella* serovars Enteritidis (Group D1 O:9), Braenderup (Group C4 O:6), Panama (Group D1 O:9), and Agona (Group B O:4), all serovars of public health importance [6]. The host range of *Salmonella* serovars demonstrated by this phage was narrow relative to other phages isolated at the same time from backyard poultry production systems. However, this narrow host range in vitro does not determine the efficacy of a phage for application in biocontrol [38], and the ability to lyse *S. Enteritidis*, the most common *Salmonella* serovar, is relevant for biocontrol purposes; additionally, a phage specific to this important serovar could represent a promising precision tool. The one-step growth curve analysis of *Salmonella* phage STGO-35-1 under standard growth conditions indicated a burst size of 122 (±10) viral particles with a latency period of 30 min (±10 min) (Figure 1). Parameters such as burst size and latency period have been previously described as relevant to characterize the lytic capacity of a phage [70]. TEM analysis showed a capsid and long flexible tail (Figure 2).

Transduction efficiencies were tested to rule out whether this phage could transduce resistance gene to a significant level, which would have been a key obstacle against its use as a biocontrol agent. The transduction efficiency studies demonstrated that STGO-35-1 is not capable of transducing an antimicrobial resistance gene, with tested volumes of 1, 5, and 20 µL of the phage at a concentration of 10^8^ PFU/mL (Appendix A). As for the positive control (phage P22 HT int), antibiotic-resistant colonies (*km*^r^) were observed, which increased in number along with the volume of phage transducer used. The frequencies of transduction did vary for the control phage from 3 × 10^−5^ to 2 × 10^−6^ PFU/CFU. While we used an MOI that could have missed some level of transduction, and the control was a high transduction mutant, further assays to test for low levels of transduction are necessary before using phages as part of biocontrol.

### 3.2. Comparative Genomics of Phage STGO-35-1

The assembly had a length of 47,483 bp, G + C content of 46.5%, and an average read coverage of 2045× (Figure 3). An analysis of the packaging strategy of this phage was conducted through PhageTerm [25]. General information on the configuration and mapping showed mapping reads of 83%, while general controls were 253 of the whole genome coverage, and with the presence of preferred termini with terminal redundancy and the occurrence of partially circular permutations, which is consistent with the “Headful” strategy of packaging. The *pac* site is located in the P1 EcoRI fragment with approximately 20 bp. [25]. The headful mode of packaging *pac* is concluded when we have a single obvious terminal only on one strand [25]. The *pac* site-directed headful packaging mechanism has been described in 9NA, FSL_SP-062, FSL_SP-069, Sasha, Sergei, and Solent [33]. Other phages similar to 9NA have also been described, which correspond to P22, P1, SPP1, Sf6, and ES18 [71,72,73,74,75,76]. This mechanism consists of the sequential encapsidation of DNA from a cleavage that produces the set of redundant and permuted molecules found in the progeny phages, as described for the phage *Escherichia coli* bacteriophage P1 [30,31]. However, it should be noted that the limitation of this method is related to the protocol used to prepare the nucleic acid libraries before sequencing [25].

Eighty-nine features were identified, including eighty-eight coding sequences (CDSs) and one tRNA (Appendix A). Forty-three annotated CDSs were identified as homologs of known phage genes, including sixteen genes encoding putative proteins involved in phage structure; nine encoding DNA-associated putative protein/enzyme replication, repair, and recombination; and five genes responsible for lytic activity, in addition to the identified tRNA-Met (CAT). The presence of tRNA genes in phages has been associated with interactions between bacterial hosts, and could participate in the translational processes, but its particular role remains unknown [77]. Additionally, a positive correlation between the number of tRNA genes and genome length has previously been reported [77]. This implies that longer phage genomes (e.g., 80 kb) would have more tRNA than average-sized genomes (e.g., 35 kb). The predicted phage tail genes were organized in a module, between nucleotide positions 24,475 and 38,191. This module was flanked by a tRNA-Met gene and putative helicase-encoding gene (Figure 3).

Genes encoding putative lysis-associated putative proteins, including a lysozyme muraminidase, a lysin_SAR-endolysin (gp18), and a putative holin, were also identified. The genome has genes whose products are associated with DNA metabolism, including a putative restriction alleviation, a DNA polymerase III, a putative transcriptional regulator, a single-stranded DNA-binding putative protein, an exonuclease, a putative ATP-dependent helicase, a DNA primase, and the phage terminase (large subunit) (Figure 3). The remaining 58 CDSs encode putative proteins of unknown function (hypothetical proteins or phage proteins; see Appendix A). No genes related to a temperate lifestyle, toxin production, virulence, or antibiotic resistance were identified. BACPHLIP [24] predicted a virulent lifestyle (with 96.25% probability). It is possible to think that the low probability of a temperate lifestyle (3.45%) is explained by a low gene flow, given, for example, by the genes coding for putative transcriptional regulators, DNA-binding proteins, phage-associated recombinases, exodeoxyribonuclease VIII, and DNA helicases (CDS 43, 67, 68, 69, 72, and 73, described in Appendix A). Therefore, the combined evaluation of the experimental lifestyle assessment (Section 2.1.3), together with the bioinformatic prediction obtained by BACPHLIP, present conclusive evidence to indicate that the STGO-35-1 phage is a virulent phage infecting *Salmonella* Enteritidis [24] (Figure 3).

On the basis of average nucleotide identity (ANI), STGO-35-1 was found to be similar to 20 available siphoviral phage genomes (Appendix A). The most similar phage was *Salmonella* phage Akira, with an ANIb of 88.53% across 54.15% of aligned sequences (47.94% ANI across the entire genome) (Appendix A). Other similar phages included *Salmonella* phage D10, *Escherichia* phage C1, and *Shigella* phage DS8 (Appendix A). Through a whole-genome phylogenetic analysis of STGO-35-1 and the 20 similar phages, a tree with minimal differences of 0.06 was obtained (Figure 4); the closest group to STGO-35-1 consisted of *Salmonella* phage KFS-SE2, *Salmonella* phage Akira, and *Salmonella* phage 64795_sal3. The whole-genome alignment of the most closely related phages to STGO-35 (Figure 4) demonstrates the similarity between their genomes and the conformation in modules (capsid proteins, tail proteins, DNA metabolism proteins, and lysins).

Finally, a phylogenetic analysis of genes encoding the putative terminase of *Salmonella* phage vB_Se_STGO-35-1, which belongs to the phage terminase, a large subunit, the PBSX family, and super family cl12054 (pfam04466). This terminase showed a close relationship between the following siphoviral terminase sequences: NCBI Protein ID:DAH84866.1, *Siphoviridae* sp. ID: DAK93255.1, and *E. coli* ID: EIH4118707.1 (Appendix A). No further details were obtained on the origin of the most related terminase sequences because some of these sequences do not correspond to the complete genome of a phage.

According to the current criteria of the International Committee on Taxonomy of Viruses (ICTV), to belong to the same genus, phage genomes should have at least 50% nucleotide identity, a similar G + C%, similar tRNA numbers, and similar coding sequences. In addition, a comparison of predicted proteomes and phylogenetic analysis must be performed [56,57]. On the basis of these criteria, we conclude that the STGO-35-1 phage represents a new species belonging to the class *Caudoviricetes*, order *Caudovirales*, and with a siphoviral morphotype, and we consider it to be an unclassified siphovirus (NCBI txid196894).

Future studies should consider the criteria proposed by ICTV to evaluate all these unclassified siphoviruses [22]. Further description of new siphoviruses is necessary, as this morphotype represents a considerable portion of phage genomes available on NCBI (around 14% (60/425) (https://www.ncbi.nlm.nih.gov/labs/virus/vssi/#/, accessed on 9 March 2022).

### 3.3. Structural Proteome Analysis of Phage vB_Sen-STGO-35-1

We experimentally verified computationally predicted STGO-35-1 structural putative proteins (Appendix A). Eighteen CDSs were predicted to encode structural putative proteins (Appendix A), while 18 putative proteins were also identified via mass spectrometry with a sequence coverage between 4.91% and 40.7% (Table 1). The identified peptides correspond to five capsid putative proteins, three minor and one major capsid putative protein, one lysine putative protein, and six tail putative proteins, including the tail tube, tape measure, minor tail, tail tip protein L, putative phage tail, and the tail spike putative proteins. Finally, one hypothetical protein and one DUF5681 family protein were retrieved (Table 1).

### 3.4. Application of STGO-35-1 Phage for Control of S. Enteritidis

We first characterized the MOI and observed that this phage adsorbs rapidly, demonstrating adsorptions of MOI 0.1 (100% adsorption), MOI 1 (60.7% adsorption), MOI 10 (68% adsorption), and MOI 100 (55.7% adsorption) (Appendix A).

Later, the assay in chicken meat showed a significant reduction (*p* < 0.05) of 2.5 log_10_ of *S. Enteritidis* in chicken meat treated with phage STGO-35-1 (Figure 5). We observed that the phage concentration remained relatively stable after a first increase in day 1, with an average of 5 × 10^8^ PFU/mL (7.7 Log_10_) (Figure 5). The reduction of *S. enterica* serovar Enteritidis observed here was similar to that previously reported after phage application in chicken meat [78,79,80]. Importantly, in this assay, only one phage was tested, which can be further used in conjunction with other different phages to achieve higher reductions. Additionally, we observed that the phage tested here remains “viable” and stable for long periods of time at low storage temperatures [81]. Phage characterization was conducted at 37 °C, and the assay in chicken meat at 4°C, which was under the same conditions found at retail. While it is not clear if phage STGO-35-1 could propagate at this condition, we observed an increase in the viral titer on day 1, and our results in meat showed that phage titers were similar in the control and treated groups, since both increased on day 1. This observation may indicate an initial lysis of another host in the control group. Furthermore, the immobilization of phage and bacteria on the food surface has been described, but further research is needed to better understand the interactions of *Salmonella*-phage-food components [81], and future studies should analyze the emergence of phage-resistant mutants as well [81].

Finally, post-harvest interventions in food safety and chicken meat have been compared with the work of Hungaro et al. [82], who compared phage reductions against chemical interventions in chicken meat and found that 2% of lactic acid, as well as 100 ppm of peroxyacetic acid, only reduced 0.8 log CFU/cm^2^ each, demonstrating that phage treatment was more efficient than tested chemical interventions. Importantly, the 2.5 log reduction found here is above the reduction accomplished by current chemical interventions. Additionally, dose–response models for *Salmonella* in chicken meat have shown that ingesting 4 logs of *Salmonella* is the level that causes illness; therefore, reducing 2.5 logs in chicken meat could have a substantial public health impact that needs to be further determined and quantified [83].

### 3.5. Genome Stability

The prediction of RBPs showed that in this phage, the genes encoding tail putative proteins are in the tail module of the genome, between the nucleotide positions 24,475 and 33,389. The identified tail putative proteins corresponded to the phage tail tube (gp55), tail tape measure (gp60), phage minor tail (gp61), tail tip protein L (gp62), putative phage tail (gp64), and tail spike protein (gp65). Of these, six tail proteins were previously identified using MS/MS (Table 2). The protein sequences with the highest identity to homologs in public databases (PDB, uniProt, NCBI_CCD and SCOpe) were gp60 tail tape measure-2 (94.74%) and gp65 tail spike (93%). Gp60 was identified as an approximately 81 kDa protein, with 31.3% of LC-MS/MS sequence coverage (Table 2). This putative protein controls the tail length by blocking the tail tube polymerization, and it is probably released from the tail shaft during infection to facilitate DNA translocation into the host cell and possibly stabilized by the covering tail assembly proteins [84,85]. Gp60 is associated with the O-antigenic polysaccharide polymerization gene (*rfbD*) an endorhamnosidase-type *Salmonella* phage receptor, which has been identified in *Salmonella* as belonging to serogroups A, B, and D1 [86].

Gp65 is approximately 72 kDa in size with 35.8% of sequence coverage (Table 2), and protein BLAST analysis indicated that this tail spike was 93.4% similar to the tail spike (QAX98701.1) of unverified *Salmonella* phage Segz_1. Subsequently, HHpred [61] showed its high level of identity with the P22 tail spike putative protein (TaxId:10754/b.80.1.6) [87,88], with 100% of probability and an E-value of 4.8 × 10^−192^. The conserved domains of this putative tail spike protein were aligned by Clustal-O [64], with the P22 tail superfamily (pfam09251), and the alignment showed a distance of 0.2. The central portion (amino acids 129 to 559) are 93.22% identical to the tail spike protein of *Salmonella phage Segz_1* and the best match was with the putative conserved domains of P22 tail superfamily (pfam09251) (Appendix A). This tail spike-like protein has been associated with the adsorption process in *Salmonella* P22 (*Podoviridae*) [88] and Det7 (*Myoviridae*) phages [87], specifically in the binding of endorhamnosidase (rhamnosyl residues) of *Salmonella enterica* (MJP01973.1) and residues of the lipopolysaccharide O-antigen [87], and both RBP predictions suggest that the LPS of *S. Enteritidis* is the receptor for phage STGO-35-1.

The relation found between the tail spike putative proteins of phage STGO-35-1 and the tail spike putative proteins of phage P22 of podoviruses has previously been described (Appendix A) [88]. In such sense, a crystal structure analysis of 9NA and P22 revealed that both phages use similar tail spikes for LPS recognition. Together with the high homology present in tail spike-like proteins (gp65 in STGO-35-1) and their distinct phylogenetic origins, identified previously by Merrill et al. [26], this may support the hypothesis of plasticity of some gene-encoding products, with mechanisms of mosaicism, which could be driven by encoded recombinases, but with a low gene content flux rate in STGO-35-1 [27].

In order to obtain an experimental approximation of the plasticity of this phage, we evaluated putative protein plasticity in STGO-35-1 after exposure to *S. Enteritidis*. We selected three variants (plaque 1, 2, and 3) using lysis plaque morphology after exposure (Appendix A), and observed two conservative nonsynonymous substitutions in gp65, with frequency percentages ranging from 67.67 to 100% (Table 3). On the other hand, the mutations observed in gp60 and gp65 from the plaque 1, 2, and 3 variants (Appendix A) suggest that their encoded proteins could be acting as main RBPs and might be subject to selection based on host availability/resistance [89,90,91,92]. Furthermore, mutations in gp65 occurred at two different positions in the genome (Table 3), which is consistent with the reported observed plasticity of the P22 tail sequence [89,91]. Therefore, it is possible, from this prediction analysis, to identify the sequence plasticity of proteins that possibly function as RBPs. In addition, many mutations have been identified in the gene encoding the tail spike of *Salmonella* phage P22, which affects the folding and stability of this protein. These mutations primarily altered the amino acids located in the central domain of the tail spike putative protein, which suggests a high plasticity of this protein domain [86,93] (Table 3). Another finding that suggests genetic plasticity in STGO-35-1 is explained by the use of the same headful packaging strategy among these three phages (9NA, P22, and STGO-35-1 [33]), but with a distinct phylogenetic origin, given the modular organization of phage genomes [27].

We selected variants using lysis plaque morphology because of evidence suggesting that the change in morphology is related to the infectivity of tail phage proteins, such as T-even phages [89,90,91,92]. The variants found could suggest small adaptive changes of the phage to its host [89]. The mutations found here occurred randomly under laboratory conditions in the presence of the host, as previously described [84,85]. Additional studies have shown a change in specificity on the original RBPs [16,17,81], which could be related to overcoming bacterial resistance to phage action (bacterial receptor switching). A recently published study [17] demonstrated the in vitro evolution of phages can be used to expand the host range and limit the emergence of phage-resistant bacteria during phage-based control of *Listeria monocytogenes*.

Our study found the importance of identifying RBPs and their plasticity. In the future, determining the mutation rate of these particular genes in complex systems, such as chicken meat or other environments, is necessary to better understand *Salmonella*–phage interactions in the environments in which phages will be applied.

## 4. Conclusions

The phage described in this study, STGO-35-1, belongs to the siphoviral morphotype (formerly family *Siphoviridae*), and has been described using an approach involving comparative genomic and phenotypic tools that have been used to identify the genomic unit of diversity inside the siphoviral morphotype. In the future, this may contribute to the classification of new similar phages. In addition, the short exposure time assay allowed us to predict the sequence plasticity of proteins that possibly function as RBPs. Finally, their successful biocontrol trial in chicken meat supports the potential use of STGO-35-1 as a biocontrol agent for targeting *S. Enteritidis*.

## Figures and Tables

**Figure 1 microorganisms-10-00606-f001:**
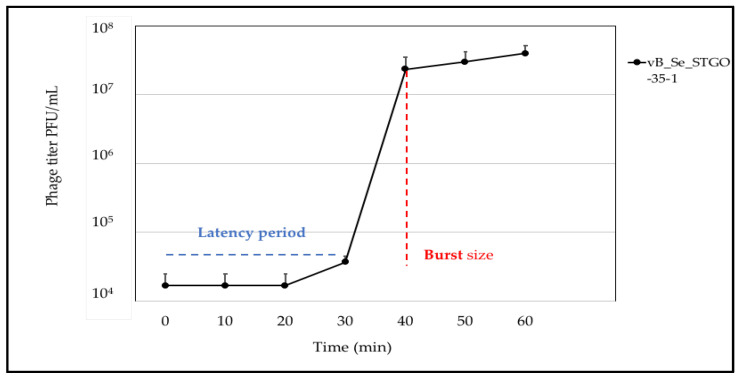
One-step growth curve of STGO-35-1. The phage concentration (PFU/mL) was followed over time after infection of *Salmonella* with phage STGO-35-1. A latency period (in blue) of 30 min (±10 min) and a burst size (in red) of 122 (±10 PFU/mL) can be observed.

**Figure 2 microorganisms-10-00606-f002:**
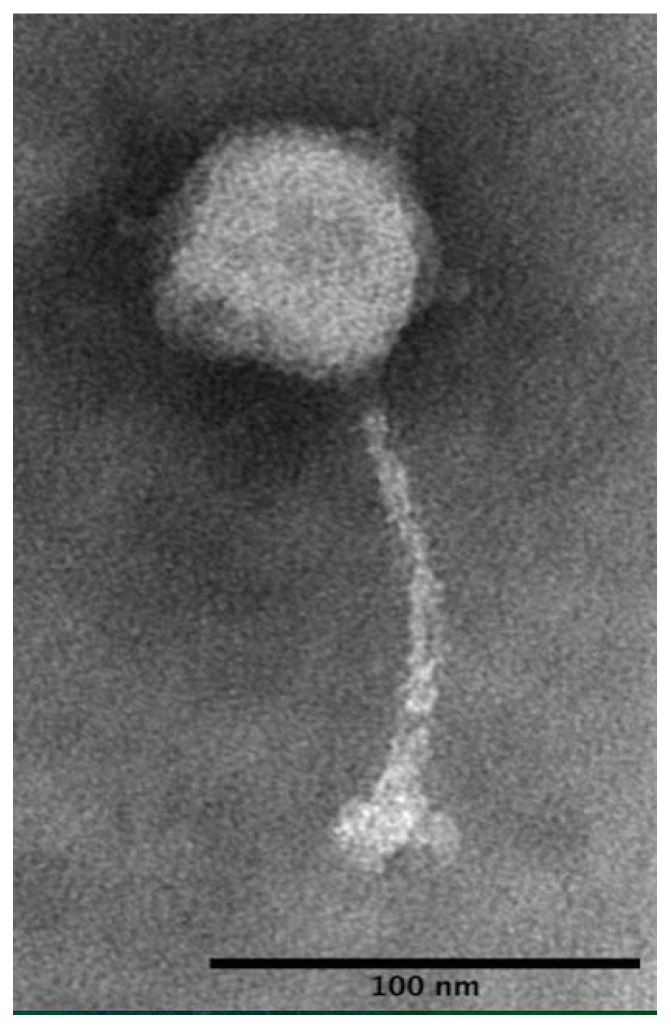
Transmission electron microscopy shows a morphology showing that this virus has a capsid, with tail and without contractile neck with siphoviral morphology. The magnification was at 100,000× in 85 kV, visualized using Fiji3 microscopy [42].

**Figure 3 microorganisms-10-00606-f003:**
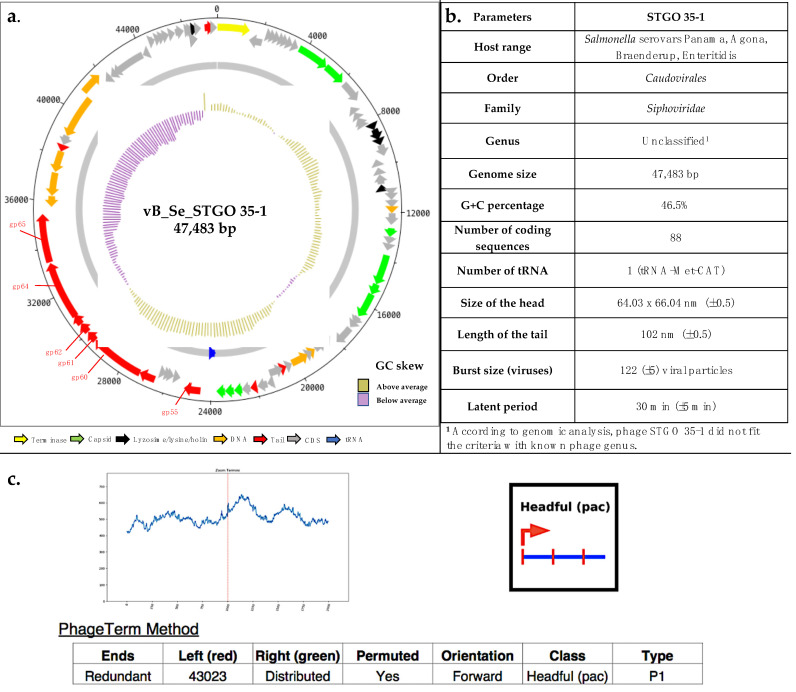
(**a**) Genome map of STGO-35-1, made with Artemis and DNAPlotter [52]. The color coding of genes indicates the functional categories of putative proteins: capsid proteins (green); lysozyme (black); CDSs and hypothetical proteins (gray); and tail proteins (red). The GC skew is represented in the inner circle and with purple indicating below average and yellow above average. (**b**) Genomic and phenotypic characterization of phage STGO 35-1. (**c**) The packaging strategies predicted in PhageTerm [25].

**Figure 4 microorganisms-10-00606-f004:**
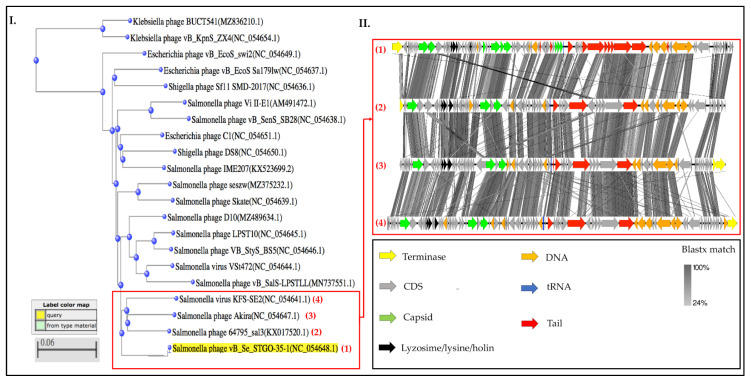
Phylogenetic relatedness of STGO-35-1 to other siphoviral morphotypes of phages. (**I**). Phylogenetic tree based on COBALT [56], using neighbor joining [55]. Phage most closely related to STGO-35-1 in red squares were marked. (**II**). Comparison of *Salmonella* phage STGO 35-1 (1) with the three most closely related phages: *Salmonella* phage KFS-SE2 (4), *Salmonella* phage Akira (3), and *Salmonella* 6795_sal3 (2). The color coding of genes indicates putative functional categories: capsid proteins (green); lysozyme (black); CDSs and hypothetical proteins (gray); tail proteins (red).

**Figure 5 microorganisms-10-00606-f005:**
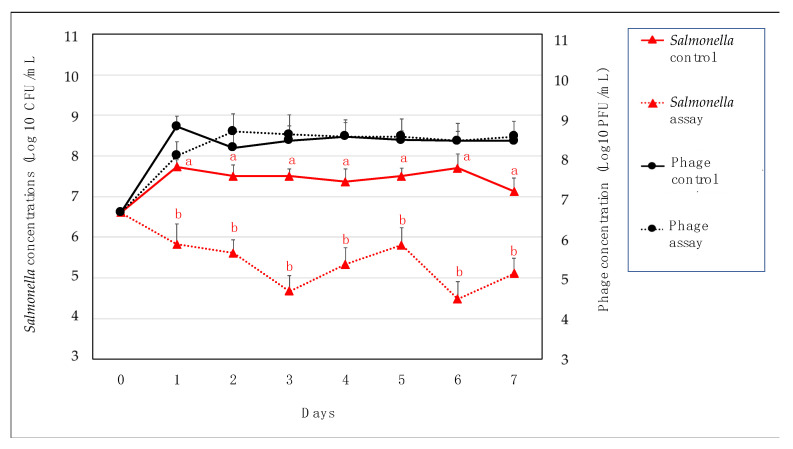
Chicken meat assay of *S. Enteritidis* and phage. This analysis represented the four experimental groups considered in this study in chicken meat pieces at a size of 2 cm^2^ approx. Experimental groups were inoculated with: (1) phage lysate at 4 × 10^6^ PFU/mL (black circle/broken lines); and (2) *Salmonella* (*SE*^nal^) at 4 × 10^6^ CFU/mL (red triangles/broken lines) (MOI = 1). (3) *Salmonella* controls (red triangles/unbroken lines) were prepared with only *Salmonella*. (4) Phage control with only phage (black circle/ unbroken lines). The standard deviation was calculated between the three replicates per day, and statistically significant differences (estimated with ANOVA at *p* < 0.05) are indicated by different letters.

**Table 1 microorganisms-10-00606-t001:** Protein homology detection and structure prediction of STGO-35-1.

	Sequence Information					HHpred Best Match		Identification via Mass Spectrometry		
	CDS	Start	Stop	Amino Acids Length	Product Name (1)	Pfam UniProtKB/Swiss-Prot ID/ NCBI ID)	Best Match Product Name	Molecular Mass (kDa)	Unique Peptide Count	Sequence Coverage (%)
	10	3690	5093	467	Capsid protein	P49859	Phage portal connector	52.52	14	31.5%
	11	5074	6030	318	Capsid protein	Q38442	Accessory head protein gp7	35.41	5	20.8%
	18	8448	8912	92	Lysin	Q6XQ98	SAR endolysin	16.86	1	11.7%
	20	9132	9620	162	DUF2514 protein	PF10721.10	Phage lysin	17.65	1	5.6%
	33	13,792	15,078	428	Capsid protein	PF03420.14/ QTH80297	Phage coat/ putative prohead core	47.22	1	4.9%
	34	15,078	15,545	155	Capsid protein	P07532.1	Capsid fiber protein	15.86	4	16.1%
	35	15,448	16,618	356	Major capsid	A0A0U5AF03	Major capsid protein	39.76	14	40.7%
	38	17,239	18,057	272	Hypothetical	NP_456098.1	Hypothetical protein	28.85	6	24.6%
	51	22,669	23,010	113	Minor capsid	PF10665.10/WP_093649519.1	Putative minor capsid	12.38	1	9.7%
	52	23,010	23,408	132	Minor capsid	PF11114.9 WP_002318693.1	Minor capsid protein	14.81	2	15.2%
	53	23,405	23,779	124	Minor capsid	PF12691.8/ DAY83110.1	TPA: minor capsid protein	14.01	1	9.7%
	55	24,475	25,191	238	Tail tube protein	PF06199.13/ WP_234600022	Phage tail tube	24.99	5	20.2%
	60	27,206	29,548	780	Tail tape measure	PF10145.10 /WP_234693434.1	Tail tape measure-2	81.26	21	31.3%
	61	29,548	30,021	157	Minor tail	PF06141/ KOX81416.1	Phage minor tail_U	18.09	6	36.3%
	62	30,021	30,491	156	Tail tip protein L	P03738.1	Tail tip protein L	17.71	2	16.0%
	64	30,887	33,349	820	Putative tail	NP_569524.1/ WP_011011097.1	Putative phage tail	91.35	9	10.4%
	65	33,389	35,416	675	Tail spike	PF09251.11/ P12528.1	P22 tail spike	72.77	19	35.8%
	88	46,936	47,355	139	DUF5681	PF18932.1	Family of (DUF5681)	15.62	2	19.4%

Putative protein detail in Appendix A. The color coding of genes indicates putative functional category, previously used in the genetic map (Figure 3), corresponding to: capsid (green); lysozyme (black); CDSs and hypothetical proteins (gray); tail proteins (red).

**Table 2 microorganisms-10-00606-t002:** Structural proteome analysis of putative receptor-binding proteins (RBPs) of the STGO-35-1 phage.

N° of CDS	GC (%)	HHPred Prediction ^1^				
		Putative Proteins Best Match	Probability	E-Value	Score	Identities (%)
55	52.4	PF06199.12 (Tail tube protein)	99.7	1.2 × 10 ^−14^	114.55	17
60	50.2	PF10145.10 (Tail tape measure-2)	100	0.0	103.18	94.74
61	51.3	PF06141 (Phage minor tail_U)	99.24	1.1 × 10 ^−10^	79.89	18
62	49.5	P03738 (Tail tip protein L)	99.69	1.2 × 10 ^−15^	112.73	8
64	47.2	NP_569524.1 (Putative phage tail)	100	3.2 × 10 ^−54^	549.04	16
65	47.9	PF09251.11 (P22 tail spike protein)	100	4.8 × 10 ^−192^	1447.29	93

^1^ Identity to homologs in public databases using HHpred [61] (PDB, uniProt, NCBI_CCD, and SCOpe).

**Table 3 microorganisms-10-00606-t003:** Information of different sub-isolates and phage selections obtained under laboratory conditions.

Parameters	Genes (N° of CDS)				
Product Name	Tape Measure (gp60)	Tail Spike (gp65)	Tail Spike (gp65)	Hypothetical Protein (gp38)	Exodeoxyribonuclease VIII (gp69)
Position	28,480	34,603	34,626	1714	37,691
Codon Change	TCT -> TCC	GAA -> GAC	ACC -> AAC	GAG -> GGG	GGA -> GGC
Effect ^1^	Synonymous Substitution	Conservative Nonsynonymous Substitution	Conservative Nonsynonymous Substitution	Radical Nonsynonymous Substitution	Synonymous Substitution
Wild-type STGO-35-1					
Ref ^2^	768	477	965	1118	976
Alt ^3^	2	432	42	0	1
Freq ^4^	0.26	47.52	4.17	0.00	0.10
Plaque 1 (P1)					
Ref ^2^	986	0	251	1401	1267
Alt ^3^	56	1318	1174	21	0
Freq ^4^	5.37	100.00	82.39	1.48	0.00
Plaque 2 (P2)					
Ref ^2^	2073	0	995	2909	2681
Alt ^3^	98	2782	2083	49	1
Freq ^4^	4.51	100.00	67.67	1.66	0.04
Plaque 3 (P3)					
Ref ^2^	2103	0	882	2790	2025
Alt ^3^	102	2762	2200	26	524
Freq ^4^	4.63	100.00	71.38	0.92	20.56

^1^ Effect based on amino acid charge and polarity—see Hanada et al. [69]; ^2^ reference allele; ^3^ alternative allele; ^4^ frequencies of mutation expressed as a percentage.

## Data Availability

The authors of this study assure that the data shared are in accordance with the consent provided by the participants on the use of confidential data.

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
