# Peer review of "Novel Salmonella Phage, vB_Sen_STGO-35-1, Characterization and Evaluation in Chicken Meat"

_microorganisms, 2022, doi:10.3390/microorganisms10030606_

Round 1

Reviewer 1 Report

The manuscript by Rivera et al. has been significantly improved. I would agree with the publication of the present version of this manuscript, but the conclusion concerning genome packaging mode seems unconvincing for me (L392-394). The circularly permuted genomes are usually packaged by a headful mechanism, not Cos, as is concluded by the authors. Moreover, I noticed that the phage STGO-35-1 encodes a large terminase, which is homologous to the PBSX family terminases (encoded by the Salmonella phage Vi II-E1 and several others) that usually support a headful genome packaging. Therefore, I would suggest the authors to check their data once more in order to make an appropriate conclusion. I think that the article of Merrill et al., 2016 (https://pubmed.ncbi.nlm.nih.gov/27561606/) would be helpful.

The authors should also check the text for possible mistakes

Author Response

Thank you very much for your comments and suggestions, we hope this version meets the requirements to be published.

Reviewer 2 Report

The manuscript includes minor revisions only.

Author Response

(The authors gave the same response as above.)

Reviewer 3 Report

Generally well written  although some grammatical errors throughout and also a tendency to use long paragraphs and long (multi-clause) sentences in some sections.

Section 3.1 Although it is referenced in the Methods section, it would be good to see detail on the host strain used - as the phenotypic observations on plaques for example may be host strain dependent.

Although host range analysis method is referenced in methods section, it would be helpful to reader to have a brief outline of approach either in results or methods section, or in a footnote to Table S1.

Section 3.2 1st paragraph. The commentary on circularly permuted genome and terminations/cos packaging is possibly over-condensed. The steps leading to these conclusions could be further described.

2nd paragraph - throughout this commentary you need to update as putative proteins/functions throughout.

Amend Table 1 so that it fits on one page maximum.

Fig S2 - figure legend to be improved - incomplete.

Fig 5 - figure legend could be improved so that it is explicit what is meant by phage/salmonella controls and assay

I agree with the conclusions drawn, although in the manuscript the authors also comment on further work that would be required to endorse use of this phage for biocontrol - some of these points should be also included in the conclusions.

Author Response

Thank you very much for your comments and suggestions, we hope this version meets the requirements to be published.

This manuscript is a resubmission of an earlier submission. The following is a list of the peer review reports and author responses from that submission.

Round 1

Reviewer 1 Report

In this manuscript, the authors describe main genomic and phenotypic characteristics of the newly isolated Salmonella siphovirus STGO-35-1 with the emphasis of those that are important for its use as a biocontrol agent. Although the data are quite extensive, they are not well described and raise a number of questions.

  1. The comments on methodology used and the analysis of the results:
  • I missed the description of the phage genome sequencing method and the explanation how did they determine that the genome is circularly permuted dsDNA. The genome sequence Accession number of this phage is not indicated;
  • The temperature at which the One-step growth experiment was performed is not indicated;
  • For phage applications, it is important to determine the temperature limits that are suitable for phage growth. Although most of the experiments were performed at 37°C, the biocontrol test was done at 4°C. It is therefore unclear whether this phage could have developed at such a temperature.
  • The poultry assay is described poorly and raises yet more questions, such as how many pieces of meat were used?; Did they use the same pieces, or different ones every day?; Sample centrifugation at 12,000 × g for 30 min seems too high for bacterial counts in the supernatant; How can they explain the increase of phage counts during first day of the control phage (without host bacteria) experiment? Why there was no an increase of Salmonella counts during the Salmonella control (without phages) experiment?
  1. The comments concerning preparation of the manuscript:
  • I missed the word ‚phage‘ or ‚virus‘ in the title of the manuscript;
  • The ‚Genome stability testing‘ would be more accurate sub-title of 2.2.3. section;
  • There should be no word ‚fibers‘ in L90 an the word ’bacteria’ in L288;
  • The reference (Figure S2) in L373 is invalid, because there are no BLASTn results of 18 phages in this Figure;
  • L379, what does the‚ capside tail‘ mean?
  • The references in the Legend of Figure 4 and in the Table S2 should be written in numbers;
  • The quality of TEM photos is not very good. I see some spikes on the capsid of these phages that might be more visible if the quality of the photos were better.
  • Phage-sensitive strains should be marked in Table S1.
  • The text of the ‚Results and discussion‘ section should be divided into more smaller sections with the emphasis on the conclusion drawn from experiments.

Reviewer 2 Report

This manuscript describes a newly-isolated phage of Salmonella enteritidis. The phage genome has been sequenced, some phage structural proteins have been characterized, and the phage has been shown to reduce bacterial viability mildly.  It has a relatively narrow host range.

Overall, the genome sequence seems to be well-annotated and its relationship to other phages is described. However, there are a number of claims and aspects of the paper that need to be addressed. In particular, the claim is made that the phage has potential for biocontrol (title), but its poor killing and narrow host range are not ideal attributes for such usage.

  1. The claims of novelty (e.g. line 316) should be tempered. The phage is clearly closely related to other phages (Fig. S2) and provides only modest new insights over these previously reported sequences.
  2. For a phage to be of utility as described, it is important that it is not temperate. It is argued that the genome does not contain genes associated with lysogeny, which is consistent with it being lytic, but genes for lysogeny can be difficult to identify. It is noteworthy that it does for example encode a Superinfection Immunity Protein (gp85). Some temperate phages form lysogens at relatively low efficiencies (1-2%), and thus the survival in Figure 5 could be explained by lysogeny.  This question needs to be resolved.
  3. Related to this, the high levels of survival are not encouraging if it is to be argued that the phage is useful for biocontrol. In fact, it suggests this phage may not be well suited for biocontrol.
  4. Similarly, it is reported that the phage has a narrow host range (line 320). However, the host range data are not shown. Table S1 shows only the strains used, not which ones are infected by the phage. Its narrow host range also argues against the general utility of the phage for biocontrol. 
  5. It is also important that for use as a biocontrol agent that the phage does not mediate generalized transduction. Transduction was not detected, but the assay has a number of limitations. First, the control used is a high transduction mutant of P22, and therefore not a fair comparison for a phage that may well do transduction at a more normal frequency.  Second, the genome is circularly permuted and likely uses a headful packaging system, and thus likely to do generalized transduction at some frequency. Third, the assay uses rather low titers of phage, and a reasonable level of transduction could be readily missed at that level of input phage.
  6. It is unclear why gene naming (numbering) is not sequential. It would be helpful to correct this.
  7. Because the genome is circularly permuted, the position for coordinate one is not predetermined. It appears to be within gene one, so that gene one is split over the ‘ends’ of the sequence file.  It would be helpful to correct this. Many similar genomes are ‘cut’ so that position one is the first base of the terminase gene.
  8. The EM’s shown in Figure 2 are of poor quality. And why are three images shown, when a single high-quality image would suffice? The legend states that the capsids have icosahedral symmetry, and this is probably true, but it cannot be deduced from these images.
  9. The protein analysis was performed by cutting bands from a gel. However, the gel image is not shown (it should be provided as supplementary data). The phage does not appear to be highly purified (e.g. CsCl banded or the like), so what is the level of confidence that the proteins are actually phage structural proteins rather than contaminating proteins (from either phage or host). For example, it is reported that the lysin protein was identified, but this is not typically a structural protein.
  10. The structural genes are annotated as being ‘capsid’ or ‘tail’. What is the justification for these assignments? Just homology?
  11. Because of the pervasive genome mosaicism of phage genomes, the whole genome phylogenies shown in Figure 4 provide few insights into the actual evolutionary relationships. If phylogenies for individual genes were determined, they would be different (because of the mosaic relationships described in the paper). Evolutionary histories cannot be simply blended.
  12. Section 3.2.2 is very poorly written, and it is difficult to understand what is being claimed. In general, the identification of plaque variants and their sequences provide little insight into gene plasticity or host range variation. This section does not evidently add to the Results and Discussion of the manuscript.

Round 2

Reviewer 1 Report

In the revised version of this manuscript, the authors made only minor changes, which are not sufficient for article publication. This manuscript still contains many inaccuracies and errors and needs a careful revision.

Comments:

  1. The description of the genome sequencing is still not provided, and the statement that “the assembled genome, possibly, consists of circularly permuted and double stranded DNA“ (L382-383) isn‘t appropriate. In the ‘Response to Reviewers‘, the authors explained that “a prediction was made with respect to phylogenetically similar genomes“, but they did not explain this to the readers. Moreover, it is unclear according which phage genomes the prediction was made. I looked at the genomes of the phages listed in the Table S4, but did not found any convincing statements that their genomes have circularly permutated genomes.
  1. The description of the poultry assay and the interpretation of the results obtained still raise questions:
  • So, what exactly the temperature of the experiment was?;
  • The authors did not explain why there is an increase of phage titre by two orders of magnitude during first day in the control phage experiment (without the host)? They state (in L282-291) that they tested for the absence of enterica in poultry meat before the experiments, but they used primers targeting tRNA gene of the STGO-35-1 phage. So, what was tested, and what the result was?;
  • L477-487; the description of the results obtained during this assay is unclear. In Figure 5, I don’t see the titres presented in the text. Another question, why there are references [58, 59] indicated next to their results.
  1. I am also not satisfied with the other corrections they made in the manuscript:
  • L92, what is “non-contractile“? Why did they delete the word “tails“?
  • L410, I still can‘t see the sequences of 18 phages in the Figure S2.
  1. The manuscript still has other mistakes that should be corrected. For example, two phages, Salmonella_Phage_64795_sal3 and Salmonella_phage_IME207 have the same accession numbers indicated in table S4.

5. In the list of authors, the same author seems to be indicated twice – in the first and the last position. Can that be the case?

Reviewer 2 Report

The authors have provided responses to the comments, and have made some revisions to the manuscript, which have helped to improve it somewhat.  However, most of the concerns (ad some of the more important ones) have not been addressed. I'm not going to repeat all the points again here.  But, for example, the response to comment 2 is that the phage does not transduce resistance genes.  I'm not sure what that means but it does not address the question and low or moderate levels of lysogeny that are difficult to deduce from plaque morphology alone.  And I don't find the argument (or the data) compelling that this phage is useful as a biocontrol agent. I understand and am very sympathetic to the constraints imposed by all of us by the pandemic.  It is especially challenging, as the pandemic complications have to be navigated to give publishable and well-supported data. The gel picture shown in the response does not look so bad that it couldn't be provided with sufficient labeling to support the protein analyses, presuming that is all correct. The reply to comment 11 does not address the question.  Ditto for comment #12.